# Optimal adaptive cancer therapy based on evolutionary game theory

**Zhiqing Li**, **Xuewen Tan**\*, **Yangtao Yu**\*

School of Mathematics and Computer Science, Yunnan Minzu University, Kunming, China

\* tanxw0910@ymu.edu.cn (XT); ynyuyyt@163.com (YY)

**Data availability statement:** All relevant data are within the paper.

**Funding:** This work was supported by the National Natural Science Foundation of China (Nos. 11361104, 12261104), the Youth Talent

## Abstract

Cancer development is a dynamic and continuously evolving process, with the emergence of drug-resistant cancer cells being one of the primary reasons for the failure of traditional treatments. Adaptive therapy, as an emerging cancer treatment strategy, is increasingly being applied in oncology. In this study, we incorporate pharmacokinetics into a cancer evolutionary game theory model and propose an optimal control problem constrained by maximum drug concentration and maximum tumor burden. Firstly, we demonstrate the existence of an optimal control for this problem. Secondly, using Pontryagin's minimum principle, we formulated the structure of the optimal control to design an optimal adaptive therapy strategy. Finally, through numerical simulations, we compare the optimal adaptive therapy strategy with other adaptive therapies and traditional treatments, and further develop personalized treatment plans for different patient groups. The results demonstrate that the optimized adaptive treatment strategy effectively preserves a high survival rate of healthy cells during treatment. By maintaining drug-sensitive and drug-resistant cell populations in a state of low-level competition, this approach prevents the proliferation of drug-resistant cells, reduces the tumor burden on patients, and extends overall survival.

## Introduction

The heterogeneity of cancer [1] and the evolution of drug resistance are significant challenges in cancer treatment. Ecological and evolutionary dynamics are increasingly recognized as crucial factors in the emergence of tumor heterogeneity and drug resistance [2–4]. Currently, maximum tolerated dose therapy and metronomic therapy [5,6] are still widely used clinical cancer treatment methods. While these treatments often show initial success, the toxic side effects of chemotherapeutic drugs and the emergence of resistance lead to the growth of uncontrolled drug-resistant cancer cell populations, making tumor recurrence inevitable.

In recent years, adaptive therapy [7,8] has emerged as an effective approach for cancer treatment. Adaptive therapy strategies involve dynamically adjusting treatment regimens based on current estimates of cancer cell growth or anticipated tumor evolution, derived from biopsies, antibody tests, and other diagnostic methods. These strategies determine the appropriate drug dosages and timing for treatment initiation or cessation. Recently, adaptive therapy has shown promising results in several clinical cancer treatments [9,10]. It is widely

Program of Xingdian Talent Support Plan (XDYC-QNRC 2022-0514), the Yunnan Provincial Basic Research Program Project (No. 202301AT070016, No. 202401AT070036), the yunnan Province International Joint Laboratory for Intelligent Integration and Application of Ethnic Multilingualism (202403AP140014). The funders had no role in study design, data collection and analysis, decision to publish, or preparation of the manuscript.

**Competing interests:** The authors declare that they have no competing interest.

acknowledged that tumor heterogeneity and the development of drug resistance result from natural selection among different cancer cell subpopulations within the tumor. Incorporating evolutionary game theory into mathematical models of cancer treatment offers a fresh viewpoint on comprehending and refining therapeutic approaches. Park and Newton [11] formulated a restricted cellular framework to explore tumor evolutionary dynamics, concentrating on random fluctuations throughout multiple adaptive chemotherapy sessions. Zhang et al. [12] proposed a pilot study where they used an evolutionary mathematical model to guide the dosage of abiraterone, aiming to postpone the emergence of resistance. Masud and Kim [13] introduced a time-dependent evolutionary dosing strategy formulated to control the persistent rivalry among drug-sensitive, resistant, and plastic cells. Bukkuri and Adler [14] developed an evolutionary game theory model to investigate the competitive interactions among cancer cells with differing biomarker production levels. They designed simulated treatment protocols aimed at driving cancer cells into evolutionary traps.

In this study, we build upon the evolutionary game theory model introduced by West et al. [15] by integrating the pharmacokinetics of chemotherapeutic drugs. Based on the constraints of the maximum drug tolerance concentration and the maximum tumor burden that the patient's body can endure, an optimal control problem is proposed. The goal of the optimal control scenario is to minimize drug toxicity and the proportion of cancer cells by the conclusion of treatment. The aim is to devise a treatment strategy that ensures the patient's body can tolerate the maximum drug concentration while simultaneously controlling and managing tumor growth so that it does not exceed the maximum tumor burden the patient can handle. Firstly, in the optimal control problem section, we show that an optimal control exists for the designed optimal control problem. Secondly, in the optimal control characterization section, We demonstrate that the state constraints for maximum drug concentration and tumor burden satisfy the regularity conditions, and we obtain the control variable values (drug dosage) at the boundary states. Using Pontryagin's minimum principle, we deduce the necessary conditions for the optimal control problem and obtain the optimal control structure. Finally, in the result section, we propose an optimal adaptive therapy scheme, comparing it with the adaptive therapy scheme proposed by West et al. and two traditional treatment strategies (MTD therapy and metronomic therapy). Additionally, we design personalized optimal adaptive therapy schemes for two types of patients: those who can tolerate high tumor burdens and those who can tolerate low tumor burdens. In the discussion section, we provide a conclusive evaluation of the optimal adaptive therapy scheme designed using the improved evolutionary game theory model, highlighting existing limitations and suggesting directions for future improvements.

## Mathematical model

West et al. [15] proposed a model of cancer progression under the strategy of harsh environment induction,which details the dynamic competitive relationships among healthy cells, drug-sensitive cancer cells, and drug-resistant cancer cells:

$$\begin{cases} \dot{x}_1 = (f_1 - \langle f \rangle)x_1, \\ \dot{x}_2 = (f_2 - \langle f \rangle)x_2, \\ \dot{x}_3 = (f_3 - \langle f \rangle)x_3, \end{cases} \tag{1}$$

where $x_1, x_2, x_3$ represent the proportions of healthy cells, drug-sensitive cells, and drug-resistant cells respectively ($x_1 + x_2 + x_3 = 1$). The function $f_i (i = 1, 2, 3)$ characterizes the fitness function of each cell subpopulation, which varies according to its environment. The term $\langle f \rangle$

denotes the average fitness of the three cell subpopulations. The fitness functions and average fitness are defined as follows:

$$f_1 = 1 - w_1 + w_1 (A\vec{x})_1 \,,$$
$$f_2 = 1 - w_2 + w_2 (A\vec{x})_2 \,,$$
$$f_3 = 1 - w_3 + w_3 (A\vec{x})_3 \,.$$

$$\langle f \rangle = f_1 x_1 + f_2 x_2 + f_3 x_3, \tag{3}$$

(2)

where $\vec{x} = (x_1, x_2, x_3)^T$, $(A\vec{x})_i$ represents the $i$-th row element of $A\vec{x}$, $w_i(i=1,2,3)$ represents the natural selection pressure experienced by each subpopulation in the tumor microenvironment, and $0 \le w_i(i=1,2,3) \le 1$. The natural selection pressure parameters for each cell subpopulation are equal before the start of chemotherapy ($w_1 = w_2 = w_3 = w_0$). $A$ represents a matrix denoting the payoff matrix of three cell subpopulations in an evolutionary game.

$$w_1 = (1+c)w_0,$$
$$w_2 = (1-c)w_0,$$
$$w_3 = w_0. \tag{4}$$

The payoff matrix for the evolutionary game is expressed as follows:

$$A = \begin{matrix} & \begin{matrix} H & S & R \end{matrix} \\ \begin{matrix} H \\ S \\ R \end{matrix} & \begin{pmatrix} a & b & d \\ e & g & h \\ k & l & o \end{pmatrix} \end{matrix},$$

where healthy cells (H), drug-sensitive cells (S), and drug-resistant cells (R) compete in pairs. In the game between healthy cells and cancer cells, healthy cells act as cooperators, while drug-sensitive and drug-resistant cells act as defectors. The game involving drug-sensitive and drug-resistant cells sees the former as defectors and the latter as cooperators. The payoffs obtained by the three cell subpopulations in their pairwise games correspond to the values in the payoff matrix. To ensure Gompertzian growth of the tumor cell population in a drug-free environment [16], the parameters in the above payoff matrix must satisfy the following inequalities:

The inequalities for the evolutionary game between the cell subpopulations are as follows:

- Between healthy cells and drug-sensitive cancer cells: $e > a > g > b$.
- Between healthy cells and drug-resistant cancer cells: $k > a > o > d$.
- Between drug-sensitive cancer cells and drug-resistant cancer cells: $h > o > g > l$.

In cancer chemotherapy, the continuous administration of chemotherapeutic drugs involves a series of processes within the human body [17,18], including absorption, distribution, metabolism, and excretion. Drugs can be administered through intravenous injection, intramuscular injection, intraperitoneal injection, oral administration, and other methods, entering the bloodstream and reaching the tumor site either directly or indirectly. As drugs accumulate and are cleared in the body, the drug concentration dynamically changes according to the dosage. Based on the above model, we introduce pharmacokinetics into the

evolutionary game model of cancer progression.

$$\begin{cases} \dot{x}_1 = (f_1 - \langle f \rangle)x_1, \\ \dot{x}_2 = (f_2 - \langle f \rangle)x_2, \\ \dot{x}_3 = (f_3 - \langle f \rangle)x_3, \\ \dot{c} = pm - qc, \end{cases} \tag{5}$$

where $c$ represents the drug concentration in the body, $m$ represents the dosage, and $p,q$ represent the pharmacokinetic rates of drug administration and clearance, respectively.

## Optimal control problem

This section introduces an optimal control problem that includes state constraints and demonstrates the existence of the optimal control. Under the state constraint conditions, by controlling the dosage of the drug, we regulate the drug concentration in the body. This guarantees that the healthy cells remain alive and maintain a high level of health,while limiting the proliferation of cancer cells and the toxic effects of the drug.

### Optimal control problem with constraints

The control function of system (5) satisfies:

$$0 \leq m(t) \leq m_{\max} < \infty, \quad \forall t \in [0, t_f], \tag{6}$$

All control functions $m(t)$ satisfying (6) are called admissible controls. The collection of all permissible controls forms the admissible control set, denoted as

$$U = \{m \in L^{\infty}([0, t_f], \mathbb{R}) : m(t) \text{ satisfies } (10)\}. \tag{7}$$

To avoid the competitive imbalance between two types of tumor cells caused by drug toxicity and to ensure that healthy cells survive at a high fitness level, we introduce the following state constraints:

$$0 \leq c \leq c_{\max}, \tag{8}$$

$$T \leq T_{\max}, \tag{9}$$

where $c_{\max}$ is the maximum tolerable drug concentration for the patient, $T = x_2 + x_3$ is the overall fraction of cancer cells, and $T_{\max}$ is the maximum tolerable tumor burden[20].

We put forward the following goal function for the control issue:

$$J(x, m) = \lambda \psi(T(t_f)) + \int_0^{t_f} \eta m(t) \, dt, \tag{10}$$

where the state variable is represented as $x = (x_1, x_2, x_3, c) \in \mathbb{R}^4$, and $\psi(T(t_f))$ denotes the overall fraction of cancer cells upon the completion of the treatment cycle, $\lambda$ and $\eta$ denote the weight coefficients.

## Existence proof of the optimal control

To facilitate understanding, we restate the optimal control problem:

$$\min J(x,m) = \lambda \psi(T(t_f)) + \int_0^{t_f} \eta m(t)\, dt, \tag{11}$$

Subject to:

$$\dot{x}(t) = f(x_1, x_2, x_3, c, m), \quad t \in [0, t_f], \tag{12}$$

$$x(0) = x_0, \tag{13}$$

State constraints:

$$0 \leq c \leq c_{\max}, \tag{14}$$

$$T \leq T_{\max}, \tag{15}$$

Control constraints:

$$0 \leq m(t) \leq m_{\max} < \infty, \quad t \in [0, t_f], \tag{16}$$

where $x_0 = (x_1^0, x_2^0, x_3^0, c^0)$.

Next, we establish that an optimal control exists [19,20]. The optimal control problem Eq (11)-(16) has an optimal solution

$$(x^*, m^*) \in W^{1,\infty}([0, t_f], \mathbb{R}^4) \times L^\infty([0, t_f], \mathbb{R})$$

such that

$$J(m^*) = \min\{J(m) : m \in U\}.$$

To establish that an optimal solution $(x^*, m^*)$ exists, these conditions must be satisfied:
1. There exists a pair $(x^*, m^*)$.
2. $G(x, U, t)$ is convex for each $(x, t)$, where

$$G(x, U, t) = \{(\eta m(t) + r, f(x, m, t)) : r \leq 0, m \in U\}. \tag{17}$$

3. The set $U$ is bounded.
4. For all $t \in [0, t_f]$ and for all feasible solutions, there exists a positive number $\delta$ such that $\|x(t)\| \leq \delta$.

Now, to prove the satisfaction of the first condition: To demonstrate the existence of a feasible solution for the optimal control problem, the following conclusion needs to be established. references [17,21,22]:

(i) $f(\cdot, m)$ is continuous for every $m \in U$.
(ii) Positive constants $N_1$ and $N_2$ exist such that for every $(x', x, m) \in (\mathbb{R}_+^4)^2 \times U$:

$$|f(x, m)| \leq N_1(1 + |x| + |m|), \tag{18}$$

$$|f(x', m) - f(x, m)| \leq N_2|x' - x|(1 + |x| + |m|). \tag{19}$$

(iii) $U$ is not empty.

Based on the state equation, it is clear that $f(\cdot, m)$ is continuous for all $m \in U$, satisfying (i). We further consider (ii):

$$
\begin{aligned}
|f(x, m)| &\leq |\dot{x}_1| + |\dot{x}_2| + |\dot{x}_3| \\
&\leq |f_1 x_1| + |f_1 x_1^2| + |f_2 x_1 x_2| + |f_3 x_1 x_3| + |f_2 x_2| + |f_1 x_1 x_2| + |f_2 x_2^2| \\
&\quad + |f_3 x_2 x_3| + |f_3 x_3| + |f_1 x_1 x_3| + |f_2 x_2 x_3| + |f_3 x_3^2| + |pm| + |qc| \\
&\leq f_1(x_1 + x_1^2 + x_1 x_2 + x_1 x_3) + f_2(x_2 + x_1 x_2 + x_2^2 + x_2 x_3) \\
&\quad + f_3(x_3 + x_1 x_3 + x_2 x_3 + x_3^2) + pm + qc \\
&= f_1 x_1(1 + x_1 + x_2 + x_3) + f_2 x_2(1 + x_1 + x_2 + x_3) \\
&\quad + f_3 x_3(1 + x_1 + x_2 + x_3) + pm + qc \\
&= 2(f_1 x_1 + f_2 x_2 + f_3 x_3) + pm + qc \\
&= 2\big[x_1 + (ax_1 + bx_2 + dx_3 - 1)(1 + c)w_0 x_1 + x_2 \\
&\quad + (ex_1 + gx_2 + hx_3 - 1)(1 - c)w_0 x_2 \\
&\quad + x_3 + (kx_1 + lx_2 + ox_3 - 1)w_0 x_3\big] + pm + qc \\
&\leq 2 + 2w_0(a + b + d - 1)(1 + c_{\max})|x_1| \\
&\quad + 2w_0(e + g + h - 1)(1 + c_{\max})|x_2| \\
&\quad + 2w_0(k + l + o - 1)|x_3| + p|m| + q|c| \\
&\leq N_1(1 + |x| + |m|),
\end{aligned}
$$

where

$$
N_1 = \max\{2,\ 2w_0(a + b + d - 1)(1 + c_{\max}),\ 2w_0(e + g + h - 1)(1 + c_{\max}),\\
2w_0(k + l + o - 1),\ p,\ q\}.
$$

Given $x' = (x_1', x_2', x_3', c')$, $x = (x_1, x_2, x_3, c) \in (\mathbb{R}_+^4)^2$, $m \in U$, we have

$$
\begin{aligned}
|f(x', m) - f(x, m)| &\leq |\dot{x}_1' - \dot{x}_1| + |\dot{x}_2' - \dot{x}_2| + |\dot{x}_3' - \dot{x}_3| + |\dot{c}' - \dot{c}| \\
&\leq |x_1' - x_1|\big[w_0(5a + 6b + 7d + 5e + g + h + 3k + l + o + 7) + 5 + 2a\big] \\
&\quad + |x_2' - x_2|\big[w_0(a + 6b + 2d + 5e + 7g + 7h + k + 4l + o) + 4\big] \\
&\quad + |x_3' - x_3|\big[w_0(2a + 2b + 7d + 2e + 2g + 7h + 3k + 4l + 4o + 8) + 6\big] \\
&\quad + |c' - c|\big[w_0(6 + a + b + 4d + e + 3g + 3h) + 2a + q\big] \\
&\leq N_2 |x' - x| \\
&\leq N_2(1 + |x' - x| + |m|),
\end{aligned}
$$

where

$$
N_2 = \max\{A, B, C, D\},
$$
$$
A = w_0(5a + 6b + 7d + 5e + g + h + 3k + l + o + 7) + 5 + 2a,
$$
$$
B = w_0(a + 6b + 2d + 5e + 7g + 7h + k + 4l + o) + 4,
$$
$$
C = w_0(2a + 2b + 7d + 2e + 2g + 7h + 3k + 4l + 4o + 8) + 6,
$$
$$
D = w_0(6 + a + b + 4d + e + 3g + 3h) + 2a + q.
$$

In summary, (ii) holds. Obviously, $U$ is non-empty, (iii) holds. Therefore, the first condition optimal control problem exists and feasible solutions exist.

Next, we prove the second condition, refer to [23]:

Let

$$\alpha_1 = (\eta m_1 + \gamma_1, x_1(f_1 - \langle f \rangle), x_2(f_2 - \langle f \rangle), x_3(f_3 - \langle f \rangle), pm_1 - qc), \exists \gamma_1 \leq 0, m \in U,$$
$$\alpha_2 = (\eta m_2 + \gamma_2, x_1(f_1 - \langle f \rangle), x_2(f_2 - \langle f \rangle), x_3(f_3 - \langle f \rangle), pm_2 - qc), \exists \gamma_2 \leq 0, m \in U,$$

and $\alpha_1, \alpha_2 \in G(x, U, t)$.

To establish the convexity of $G(x,U,t)$, it is necessary to satisfy the following conditions:

$$\alpha_3 = \mu \alpha_1 + (1 - \mu)\alpha_2 \in G(x, U, t), \mu \in [0, 1].$$

Let

$$z_1 = \mu(\eta m_1 + \gamma_1) + (1 - \mu)(\eta m_2 + \gamma_2) = \mu \eta m_1 + \eta(1 - \mu)m_2 + \mu \gamma_1 + (1 - \mu)\gamma_2.$$

Thus, let $\eta m_3 = \eta(\mu m_1 + (1 - \mu)m_2), \gamma_3 = \mu \gamma_1 + (1 - \mu)\gamma_2$. Clearly, $m_3 \in [0, 1]$, and since $\gamma_1, \gamma_2 \leq 0$, we have $\gamma_3 \leq 0$.

$$z_2 = \mu(f_1 - \langle f \rangle)x_1 + (1 - \mu)(f_1 - \langle f \rangle)x_1 = (f_1 - \langle f \rangle)x_1,$$
$$z_3 = \mu(f_2 - \langle f \rangle)x_2 + (1 - \mu)(f_2 - \langle f \rangle)x_2 = (f_2 - \langle f \rangle)x_2,$$
$$z_4 = \mu(f_3 - \langle f \rangle)x_3 + (1 - \mu)(f_3 - \langle f \rangle)x_3 = (f_3 - \langle f \rangle)x_3,$$
$$z_5 = \mu(pm_1 - qc) + (1 - \mu)(pm_2 - qc) = pm_3 - qc.$$

Therefore, $\alpha_3 = \mu \alpha_1 + (1 - \mu)\alpha_2 \in G(x, U, t)$, which proves that $G(x,U,t)$ is convex for every $(x,t)$.

Additionally, since the admissible control set is defined to be bounded and closed, we can conclude that $U$ is bounded and closed. Since for all $t \in [0, t_f]$, the solutions to the optimal control problem are bounded, we can find a $\delta > 0$ for which $\|x(t)\| \leq \delta$ holds. Therefore, the third and fourth conditions are also satisfied. In summary, the existence proof of the optimal control problem is complete.

## Optimal control characterization

In this section, we use the method of Tan et al. [24] to prove that the state constraints satisfy regularity conditions and separately obtain the boundary drug dosages under each state. To explore the requisite conditions for the optimal control problem, we utilized Pontryagin's minimum principle and formulated the optimal control structure for the problem.

### State constraints

For the state constraint $0 \leq c \leq c_{\max}$, consider the state equation

$$\dot{c} = pm - qc.$$

The control variable $m$ is explicitly included in the state equation, which adheres to the first-order regularity criterion

$$\frac{\partial \dot{c}}{\partial m} = p \neq 0.$$

During continuous administration of the drug, the drug concentration reaches $c = c_{\max}$. At this boundary state, $\dot{c} = 0$, and we obtain the boundary drug dosage

$$m_1 = \frac{qc_{\max}}{p}. \tag{20}$$

For the state constraint $T \leq T_{\max}$, define

$$E(t) = T - T_{\max} = x_2(t) + x_3(t) - T_{\max}, \tag{21}$$

the boundary state $T = T_{\max}$ can be expressed as

$$
\begin{aligned}
0 = \dot{E}(t) &= \dot{x}_2(t) + \dot{x}_3(t) \\
&= (f_2 - \langle f \rangle)x_2 + (f_3 - \langle f \rangle)x_3 \\
&= f_2 x_1 x_2 - f_1 x_1(1 - x_1) + f_3 x_1 x_3 \\
&= x_1(\langle f \rangle - f_1) \\
&= -\dot{x}_1.
\end{aligned}
$$

Clearly, the control variable $m(t)$ is not directly present in the preceding equation. Next, let's examine the second-order derivative,

$$
\begin{aligned}
0 = \ddot{E}(t) &= (w_0(pm - qc)K_1 + w_0(1 + c)\dot{K}_1)(x_1^2 - x_1) \\
&\quad + (1 + w_0(1 + c)K_1)(2x_1 - 1) \\
&\quad + (w_0(qc - pm)K_2 + w_0(1 - c)\dot{K}_2)x_2 \\
&\quad + (1 + w_0(1 - c)K_2)\dot{x}_2 \\
&\quad + w_0 \dot{K}_3 x_3 + (1 + w_0 K_3)\dot{x}_3 \\
&= w_0 p(K_1(x_1^2 - x_1) - K_2 x_2)m + L_1 + L_2,
\end{aligned}
$$

where

$$
\begin{aligned}
K_1 &= ax_1 + bx_2 + dx_3 - 1, \\
K_2 &= ex_1 + gx_2 + hx_3 - 1, \\
K_3 &= kx_1 + lx_2 + ox_3 - 1,
\end{aligned}
$$

$$
\begin{aligned}
L_1 &= w_0(1 + c)\dot{K}_1(x_1^2 - x_1) + (1 + w_0(1 + c)K_1)(2x_1 - 1) \\
&\quad + w_0(1 - c)\dot{K}_2 x_2 + (1 + w_0(1 - c)K_2)\dot{x}_2 + w_0 \dot{K}_3 x_3 + (1 + w_0 K_3)\dot{x}_3, \\
L_2 &= w_0 qc K_2 x_2 - w_0 qc K_1(x_1^2 - x_1).
\end{aligned}
$$

From the previous equations, it is evident that the control variable $m$ first emerges in the second derivative of $E(t)$. Hence, a second-order state constraint is present.

Since

$$\frac{\partial \ddot{E}(t)}{\partial m} = w_0 p(K_1(x_1^2 - x_1) - K_2 x_2) \neq 0, \tag{22}$$

the regularity condition is satisfied. Further, for the boundary control $T = T_{\max}$, setting $0 = \ddot{E}(t)$, we obtain the boundary drug dosage

$$m_2 = -\frac{(L_1 + L_2)}{w_0 p(K_1(x_1^2 - x_1) - K_2 x_2)}. \tag{23}$$

## Optimal control structure

We introduce the adjoint variable $\vartheta = (\vartheta_1, \vartheta_2, \vartheta_3, \vartheta_4) \in \mathbb{R}^4$. According to Pontryagin's minimum principle [25–27], the standard Hamiltonian function is characterized as:

$$H(\vartheta(t), x(t), m(t), t) = \eta m(t) + \vartheta_1(f_1 - \langle f \rangle)x_1 + \vartheta_2(f_2 - \langle f \rangle)x_2$$
$$+ \vartheta_3(f_3 - \langle f \rangle)x_3 + \vartheta_4(pm - qc). \tag{24}$$

Furthermore, we introduce the Lagrange multipliers $\xi_1, \xi_2$ to connect the state constraints with the Hamiltonian function, obtaining the augmented Hamiltonian function:

$$H(\vartheta(t), x(t), m(t), \xi_1(t), \xi_2(t), t) = H(\vartheta(t), x(t), m(t), t) + \xi_1(c - c_{\max})$$
$$+ \xi_2(T - T_{\max}). \tag{25}$$

Should an optimal control solution $(x^*, m^*)$ be found, the subsequent adjoint equations and transversality conditions must hold:

1. Adjoint differential equations:

$$\dot{\vartheta}(t) = -\frac{\partial H(\vartheta(t), x(t), m(t), \xi_1(t), \xi_2(t), t)}{\partial x}.$$

2. Transversality condition:

$$\vartheta(t_f) = \frac{\partial \psi(t_f)}{\partial x}.$$

3. Minimum condition:

$$H(\vartheta(t), x(t), m^*(t), \xi_1(t), \xi_2(t), t) \leq H(\vartheta(t), x(t), m(t), \xi_1(t), \xi_2(t), t).$$

4. Complementary slackness condition:

$$\xi_1(t) \leq 0, \quad \xi_1(c - c_{\max}) = 0,$$
$$\xi_2(t) \leq 0, \quad \xi_2(T - T_{\max}) = 0.$$

Subsequently, we derive the system of adjoint equations:

$$\begin{aligned}
\dot{\vartheta}_1 &= \vartheta_1\left(\langle f \rangle - f_1 + x_1(Y_1 - aw_0(1+c))\right) \\
&\quad + \vartheta_2 x_2(Y_1 - ew_0(1-c)) + \vartheta_3 x_3(Y_1 - kw_0), \\
\dot{\vartheta}_2 &= \vartheta_1 x_1(Y_2 - bw_0(1+c)) \\
&\quad + \vartheta_2\left(\langle f \rangle - f_2 + x_2(Y_2 - gw_0(1-c))\right) + \vartheta_3 x_3(Y_2 - lw_0) - \xi_2, \\
\dot{\vartheta}_3 &= \vartheta_1 x_1(Y_3 - dw_0(1+c)) \\
&\quad + \vartheta_2 x_2(Y_3 - hw_0(1-c)) + \vartheta_3\left(\langle f \rangle - f_3 + x_3(Y_3 - ow_0)\right) - \xi_2, \\
\dot{\vartheta}_4 &= \vartheta_1 x_1\left(w_0 K_1 x_1 - w_0 K_2 x_2 - w_0 K_1\right) \\
&\quad + \vartheta_2 x_2\left(w_0 K_1 x_1 - w_0 K_2 x_2 - w_0 K_2\right) + \vartheta_3 x_3\left(w_0 K_1 x_1 - w_0 K_2 x_2\right) - \xi_1,
\end{aligned} \tag{26}$$

To facilitate the discussion, the terms $\langle f \rangle$ and $f_i$ in the adjoint equation system, where $i = 1, 2, 3$, will not be expanded, and

$$Y_1 = aw_0(1+c)x_1 + ew_0(1-c)x_2 + kw_0x_3 + f_1,$$
$$Y_2 = bw_0(1+c)x_1 + gw_0(1-c)x_2 + lw_0x_3 + f_2,$$
$$Y_3 = dw_0(1+c)x_1 + hw_0(1-c)x_2 + ow_0x_3 + f_3.$$

Based on Pontryagin's minimum principle, the optimal control structure is derived as follows:

$$\frac{\partial H(\vartheta(t), x(t), m(t), \xi_1(t), \xi_2(t), t)}{\partial m} = 0.$$

We get the switching function:

$$\sigma(t) = \eta + p\vartheta_4. \tag{27}$$

Consequently, the optimal control structure is represented as follows:

$$m(t) = \begin{cases} 1 & \sigma(t) < 0, \\ m_i & \sigma(t) = 0, \\ 0 & \sigma(t) > 0. \end{cases} \tag{28}$$

Based on the optimal control structure obtained from Pontryagin's minimum principle, we propose the optimal adaptive therapy for the cancer evolutionary game model. For patients with different initial tumor cell proportions, the core idea of optimal adaptive therapy is to maintain a high survival proportion of normal cells, ensuring a healthy competition between drug-sensitive and drug-resistant cells within the patient or tumor burden (maximum tumor burden), and ensuring the patient's overall fitness while maintaining the drug concentration below the maximum tolerable level. The timing of administration and the dosage of the drug are determined based on the real-time detected proportions of the two cancer cell subpopulations.

During the initial phase of treatment, we utilize the highest tolerable dose of the medication to ensure that healthy cells maintain a higher fitness and competitive edge over tumor cells. This approach leads to a swift reduction in drug-sensitive cancer cells. When the drug concentration within the patient's system reaches the highest tolerable level, we cease administering the medication.At this point, the duration of drug administration is relatively short, and the competitive ability of the drug-resistant cells is not yet fully manifested. However, the drug-sensitive cells will rapidly grow once the drug is stopped, while the healthy cells maintain relatively high fitness. By sustaining this periodic method of drug administration, we progress to the next phase of treatment when the overall tumor cell proportion in the patient's body hits the peak tumor burden. We then resume drug administration until its level in the patient's system is back to the highest tolerable limit, ensuring an equilibrium between drug-sensitive and drug-resistant cells. This strategy prevents the complete eradication of drug-sensitive cells and maintains competitive pressure between the two cell types. When the total proportion of tumor cells reaches a relatively stable state, we continue to administer the drug at the lowest effective level. During the second treatment phase, according to the cyclic treatment method, the drug administration is cyclically paused or resumed at the optimal time intervals. This ensures that the normal cells maintain a high survival rate, and the two types of cancer cells compete effectively within the patient's body. This strategy prevents the predominance of drug-resistant cells and minimizes the negative impacts of the medication on the patient's health, thereby extending survival time and enhancing therapeutic outcomes.

In this paper, we consider four treatment cycles and provide the optimal control structure as follows:

$$
m(t) = \begin{cases}
1, & 0 \leq t \leq t_1, \\
m_1, & t_1 \leq t \leq t_2, \\
0, & t_2 \leq t \leq t_3, \\
m_2, & t_3 \leq t \leq t_4, \\
0, & t_4 \leq t \leq t_5, \\
m_3, & t_5 \leq t \leq t_6, \\
0, & t_6 \leq t \leq t_7, \\
m_4, & t_7 \leq t \leq t_8, \\
0, & t_8 \leq t \leq t_f.
\end{cases}
\tag{29}
$$

## Result

In this section, we employ numerical simulations to quantitatively optimize the timing and dosage of the adaptive treatment scheme. To address the optimal control problems Eq (11)-(16), we apply a discretization method. For simulating other therapeutic schemes, we utilize the ode45 solver in MATLAB 2019.We use the parameters from reference [11] and made some adjustments to the parameters of the benefit matrix due to environmental factors for the numerical simulations. The parameters used for the numerical simulations are shown in Table 1.

### Continuous drug treatment simulation

To understand how optimal adaptive therapy strategies sustain healthy competition among different cell subpopulations and achieve sustainable treatment, we first study the dynamic evolution of the prisoner's dilemma cancer evolution game model both without treatment and under various conventional treatment schemes.

Fig 1 illustrates the dynamic changes in the proportions of each cell type under no treatment conditions for cases with Fig 1 (A) high tumor proportion, Fig 1 (B) medium tumor proportion, and Fig 1 (C) low tumor proportion. Regardless of the initial tumor proportion, in the absence of chemotherapeutic intervention, the three cell subpopulations naturally engage in a prisoner's dilemma game. The growth or decline of each cell subpopulation entirely depends on the payoff parameters in the payoff matrix. In all pairwise games between cell subpopulations, drug-sensitive cells always act as defectors, thus showing a natural advantage in the evolutionary process until they dominate the entire cell population.

Fig 2 describes the Maximum Tolerated Dose (MTD) treatment scheme. After 25 days of continuous treatment with the maximum drug dose, Fig 2 (A) the healthy cell population emerges as the dominant group upon completion of the treatment. However, the substantial

**Table 1. Parameter and value Table [11].**

| Parameter | $a$ | $b$ | $d$ | $e$ | $g$ | $h$ | $k$ |
|---|---|---|---|---|---|---|---|
| Value | 3 | 1.5 | 1.5 | 4 | 2 | 2.8 | 3.7 |
| Parameter | $l$ | $o$ | $p$ | $q$ | $c_{max}$ | $T_{max}$ | $w_0$ |
| Value | 1 | 2.2 | 1.2 | 1.5 | 0.7 | 0.5 | 0.2 |

The values of $c_{max}$ and $T_{max}$ can be adjusted in accordance with the physical conditions of various patient groups.

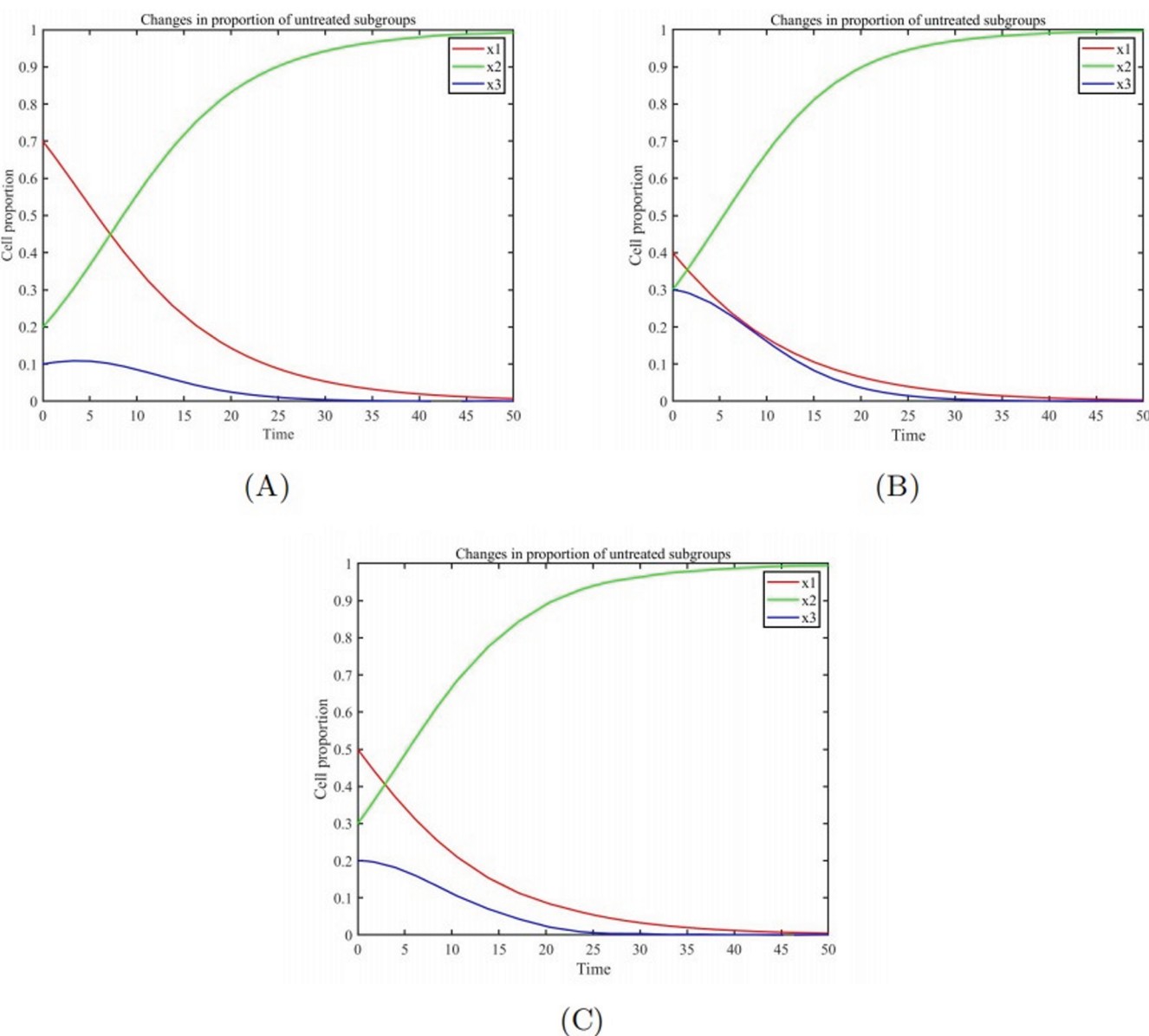

**Fig 1. Changes in the proportion of untreated subgroups.** (A) Initial low cancer cell proportion. (B) Initial high cancer cell proportion. (C) Initial medium cancer cell proportion.

drug dosage eradicates a considerable amount of drug-sensitive cells during therapy, causing them to forfeit their competitive advantage over drug-resistant cells. As a result, drug-resistant cells rapidly become dominant once the treatment is halted. Subsequently, as the drug concentration in the patient's body gradually diminishes, drug-sensitive cells regain their competitive advantage and reestablish dominance in the cell population.However, regardless of which cancer cell population becomes dominant, it signifies a failure of the treatment, as evidenced by the changes in the total cancer cell proportion depicted in Fig 2 (B). Fig 3 depicts the approach of ongoing low-dose therapy. It becomes evident that continuously administering a low drug dose to avert resistance development is impractical. In Fig 3 (A),

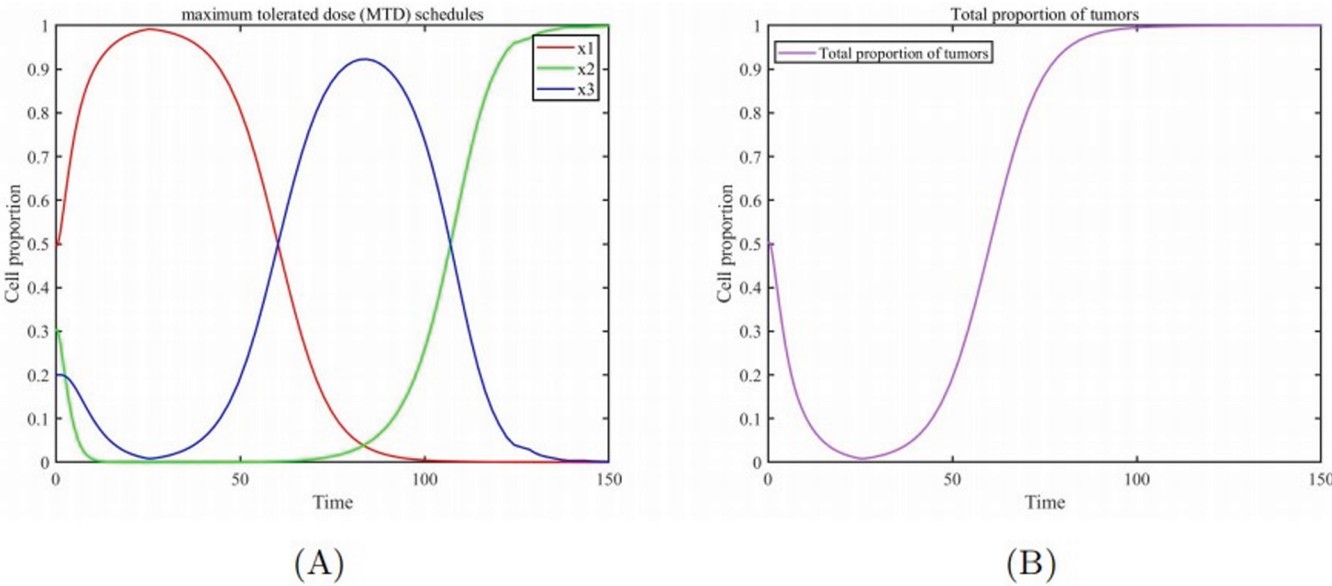

**Fig 2. Maximum tolerated dose (MTD) treatment scheme.** (A) Temporal evolution of various cell subpopulation proportions during 25 days of continuous treatment at the maximum tolerated dose and following cessation of treatment. (B) Temporal evolution of the overall proportion of cancer cells.

after a brief period where healthy cells and drug-resistant cells gain a competitive advantage, they are eventually replaced by drug-sensitive cells. In Fig 3 (B), the overall proportion of cancer cells shows a slight reduction initially but gradually dominates the entire cell population again. This is because continuous low-dose treatment fails to suppress the competitive advantage of drug-sensitive cells, highlighting the need for an appropriate drug dosage in cancer treatment. Fig 4 illustrates the metronomic therapy strategy. In Fig 4(A), due to the fixed drug dosage and fixed treatment on/off times, this strategy fails to maintain the competitive balance among the three cell subpopulations. Healthy cells initially maintain a high proportion but are soon replaced by drug-sensitive cells. In Fig 4 (B), the overall proportion of cancer cells fluctuates under intermittent drug administration, ultimately dominating the entire cell population. The metronomic therapy strategy highlights that appropriate timing for treatment on/off cycles is crucial for successful therapy.

The adaptive therapy strategy developed by Newton and Ma [28] involves selecting appropriate drug dosages to enable the three cell subpopulations to cycle through a fixed-ratio competition. Essentially, the drug dosage, treatment duration, and treatment intervals are fixed. Considering the importance of drug dosages and treatment timing, we propose the optimal adaptive therapy strategy.

## Optimize treatment simulation

In Figs 5 and 6, we describe cases of high tumor proportion treated with the optimal adaptive therapy strategy. The proportions of healthy cells, drug-resistant cells, and drug-sensitive cells are $x_1$, $x_2$, and $x_3 = (0.4, 0.05, 0.01)$. Fig 5 (A) shows the drug dosages and the treatment on/off periods during the optimal adaptive therapy process.Fig 5 (B) illustrates the changes in drug concentration within the patient over time. Because of the pharmacokinetics of drug clearance and accumulation, the drug concentration varies periodically. Through numerical simulation, we obtained the optimal drug dosages ($m_1 = 0.875, m_2 = 0.8745, m_3 = 0.8745,$

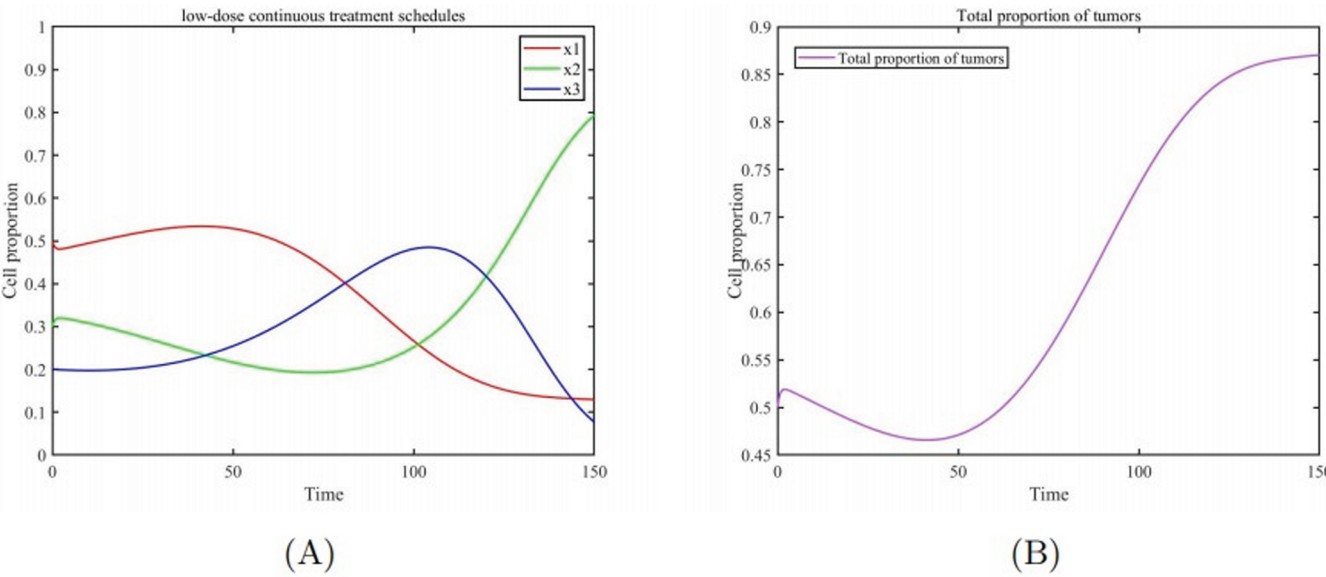

**Fig 3. Low-dose continuous treatment (drug dose m=0.352).** (A) Temporal evolution of the proportions of various cell subpopulations. (B) Temporal evolution of the overall proportion of cancer cells.

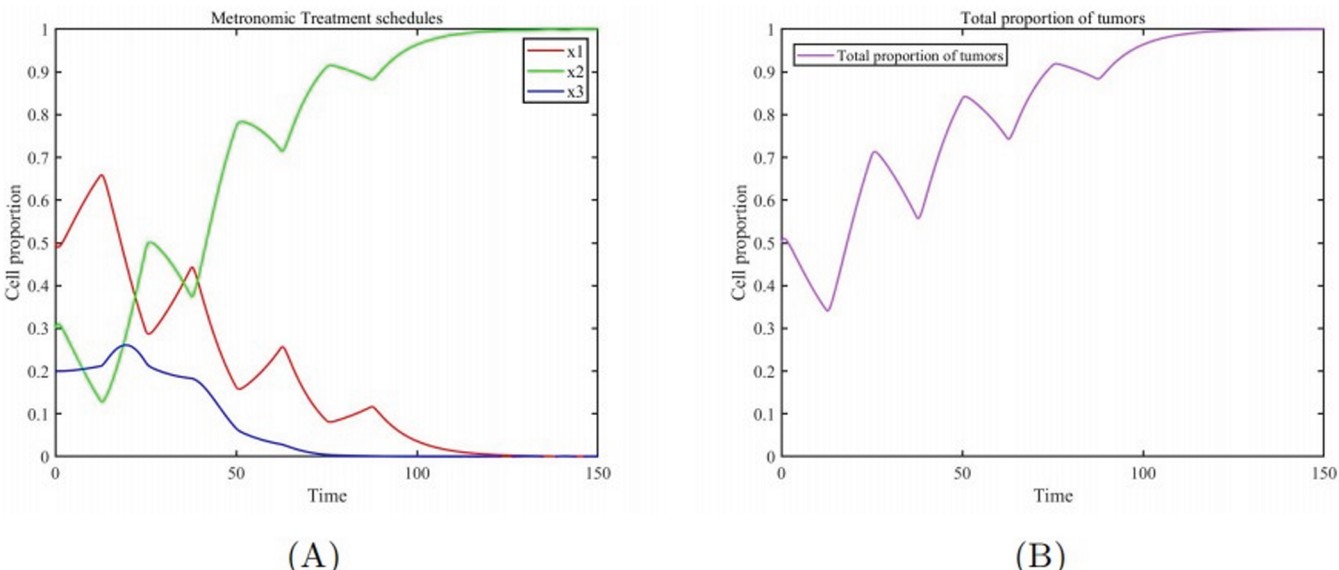

**Fig 4. Metronomic therapy (drug dose m=0.5, treatment interval of 12.5 days).** (A) Temporal evolution of the proportions of various cell subpopulations. (B) Temporal evolution of the overall proportion of cancer cells.

$m_4 = 0.8745$) and the treatment periods ($t_1 = 1, t_2 = 6.2, t_3 = 14.2, t_4 = 18.6, t_5 = 27, t_6 = 31.6, t_7 = 39.6, t_8 = 44.8$). For cases with a high proportion of cancer cells, treatment starts with a short-term maximum drug dosage to eliminate some drug-sensitive cells. The therapy then switches to a different dosage until the drug concentration within the patient attains the maximum tolerable level($c_{max} = 0.7$), at which point drug administration is halted. At this point,

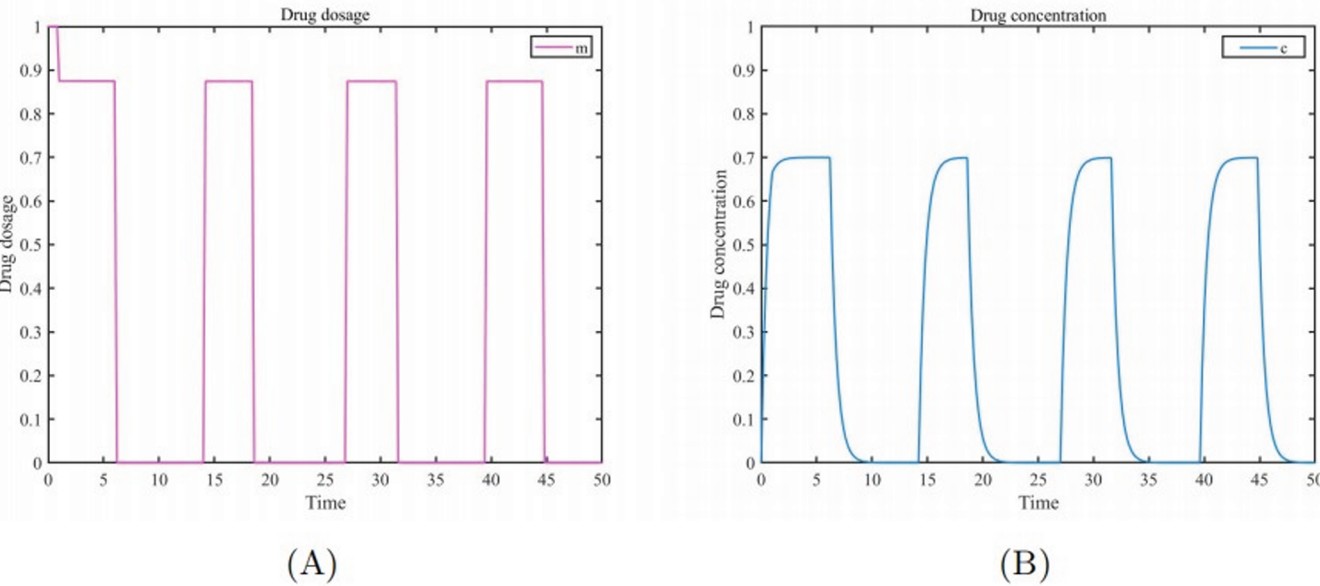

**Fig 5. (A) Drug dosage.** (B) Drug concentration changes.

healthy cells transition from a low to a high survival proportion, maintaining a balanced competition with drug-sensitive and drug-resistant cells.When the tumor burden in the patient's body increases to the maximum tolerable level ($T_{max}$ = 0.45), treatment is resumed until the drug concentration returns to its peak level, initiating the second treatment period. According to the treatment steps, Fig 6 (A) illustrates the cyclic treatment based on the calculated optimal dosage and best on/off times, which intervenes in the competitive interactions among the three subpopulations. Healthy cells increase to a high proportion and then maintain this level with some fluctuations. Both drug-sensitive and drug-resistant cells retain their competitive abilities under the influence of chemotherapy, ensuring that both cancer cell subpopulations remain within the patient's acceptable tumor threshold. Fig 6 (B) shows that the overall level of cancer cells fluctuates at a low proportion throughout the treatment process.

Compared to the adaptive therapy strategy proposed by Newton and Ma [28], the optimal adaptive therapy strategy presented here takes into account the patient's drug toxicity tolerance and the tolerable tumor burden range, adjusting drug dosages and treatment on/off times accordingly. Due to the growth dynamics of cancer cells, the proliferation period and drug effect window vary. By precisely controlling drug dosages and treatment times, it is possible to intervene during the most drug-sensitive phases of tumor cell growth, thereby enhancing treatment efficacy and avoiding resistance development. For cases with a high initial tumor burden, the adaptive therapy strategy proposed by West et al.disregards the patient's well-being throughout the treatment. As the tumor cell levels in the patient's body continuously cycle back to the initial tumor state due to the closed-loop design of the treatment plan, the patient's condition deteriorates repeatedly. Over time, the patient's ability to endure high tumor burdens may diminish due to repeated stress, ultimately resulting in treatment failure. In the aforementioned simulation results, the optimal adaptive therapy strategy successfully treated cases with a high initial tumor burden. Healthy cells became the dominant population, while drug-sensitive and drug-resistant cells maintained stable competition at low threshold levels. This approach prevented patients from repeatedly enduring high

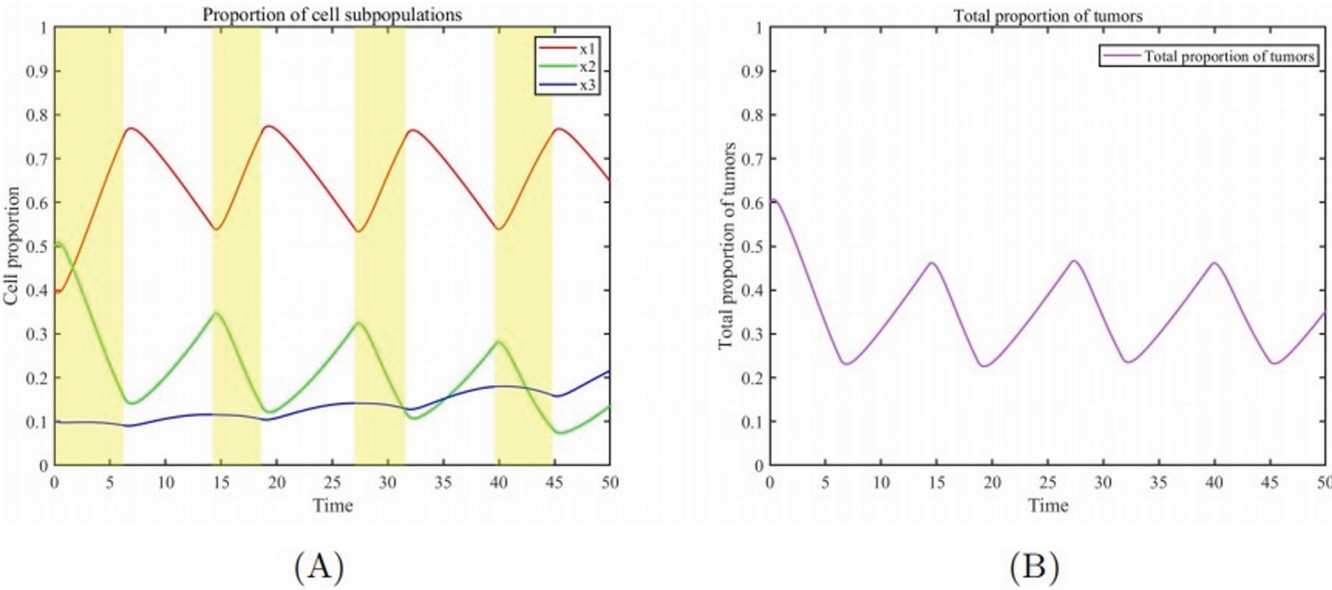

**Fig 6. Optimal adaptive therapy.** (A) Temporal evolution of the proportions of various cell subpopulations, Pale yellow and white backgrounds respectively indicate the treatment phase and the treatment interval phase (This will not be repeated further in the text.). (B) Temporal evolution of the overall proportion of cancer cells.

tumor burdens during treatment, minimized the toxic side effects of the drugs, and ensured a high quality of life for patients throughout the treatment process.

## Comparison of treatment options

Next, we compare the treatment effectiveness of the optimal adaptive therapy strategy with some traditional cancer treatment methods under the same initial proportions of cancer cells.

We consider the treatment simulation for cases with initial proportions of healthy cells and cancer cells $x_1, x_2, x_3 = (0.6, 0.3, 0.1)$ using three different treatment strategies with the same total drug dosage. We consider treatments over a limited time period, $t \in [0, 50]$. In Fig 7 (A),(B) and (C) represent the changes in cell proportions over time for MTD treatment, metronomic therapy, and optimal adaptive therapy, respectively. MTD treatment results in the rapid reduction of drug-sensitive cells to near extinction. Throughout the post-treatment phase, drug-resistant cells acquire a notable edge over drug-sensitive cells, resulting in a competitive proliferation of drug-resistant cells. Consequently, by the conclusion of treatment, the quantity of healthy cells in the patient's body tends to be completely surpassed by drug-resistant cells.While the metronomic therapy strategy effectively manages drug-resistant cells, during the advanced stages of treatment, drug-sensitive cells gradually obtain an advantage over healthy cells due to the failure to treat them effectively at the optimal moment. Consequently, healthy cells are gradually replaced by drug-sensitive cells. Towards the conclusion of treatment, drug-sensitive cells almost completely dominate the entire cell population. The optimal adaptive therapy persistently fine-tunes drug dosages to sustain a balanced interaction between drug-sensitive and drug-resistant cells. This method guarantees that the patient can endure the drug toxicity and tumor burden while hindering the emergence of drug resistance. Throughout the entire treatment period up to the conclusion of therapy, healthy cells maintain a high proportion, while the overall cancer cell population remains in a low-value

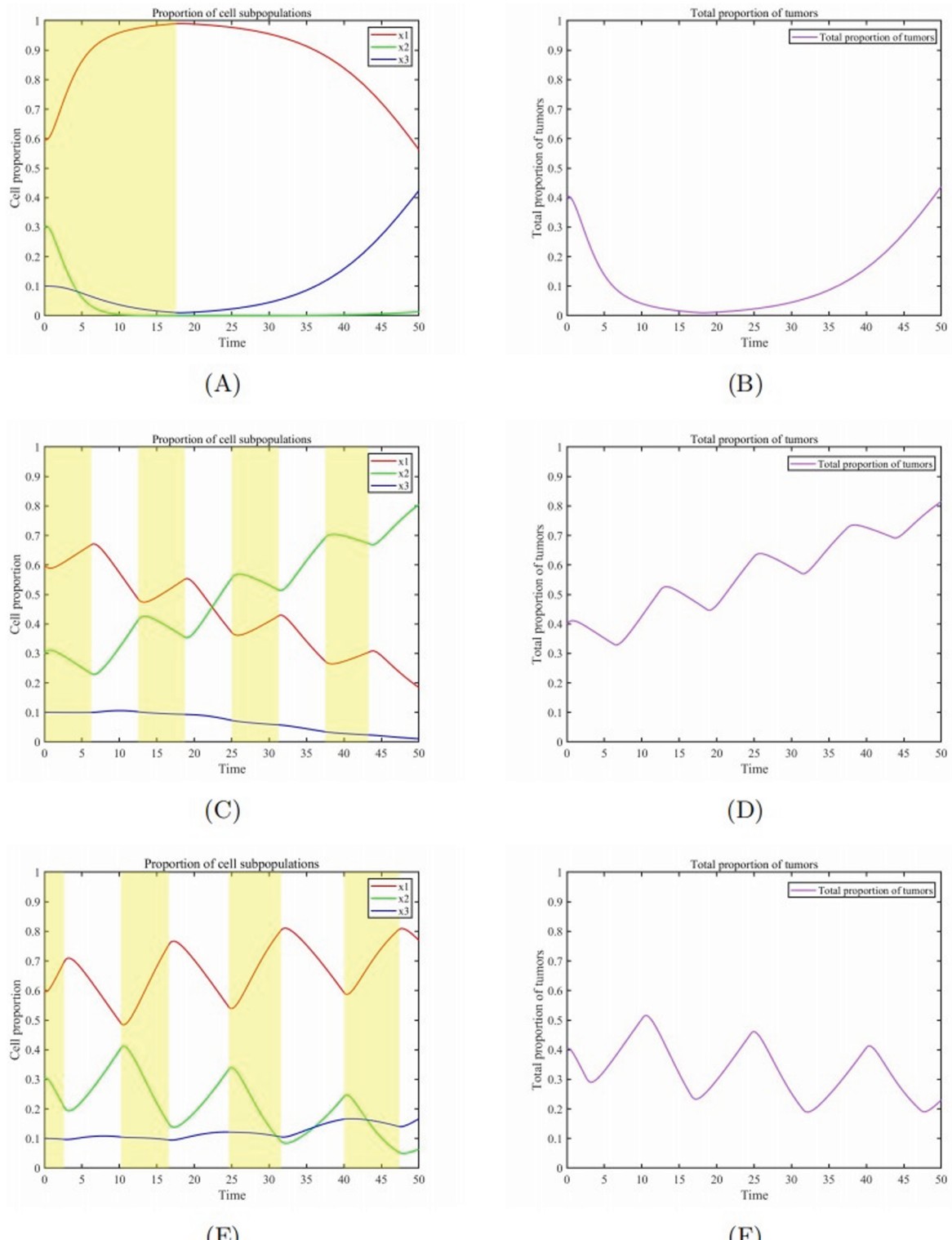

**Fig 7. Comparison of treatment plans.** (A) MTD treatment scheme:Temporal evolution of the proportions of various cell subpopulations. (B) MTD treatment scheme: Temporal evolution of the overall proportion of cancer cells. (C) Metronomic therapy scheme: Temporal evolution of the proportions of various cell subpopulations. (D) Metronomic therapy scheme: Temporal evolution of the overall proportion of cancer cells. (E) Optimal adaptive therapy scheme: Temporal evolution of the proportions of various cell subpopulations. (F) Optimal adaptive therapy scheme: Temporal evolution of the overall proportion of cancer cells.

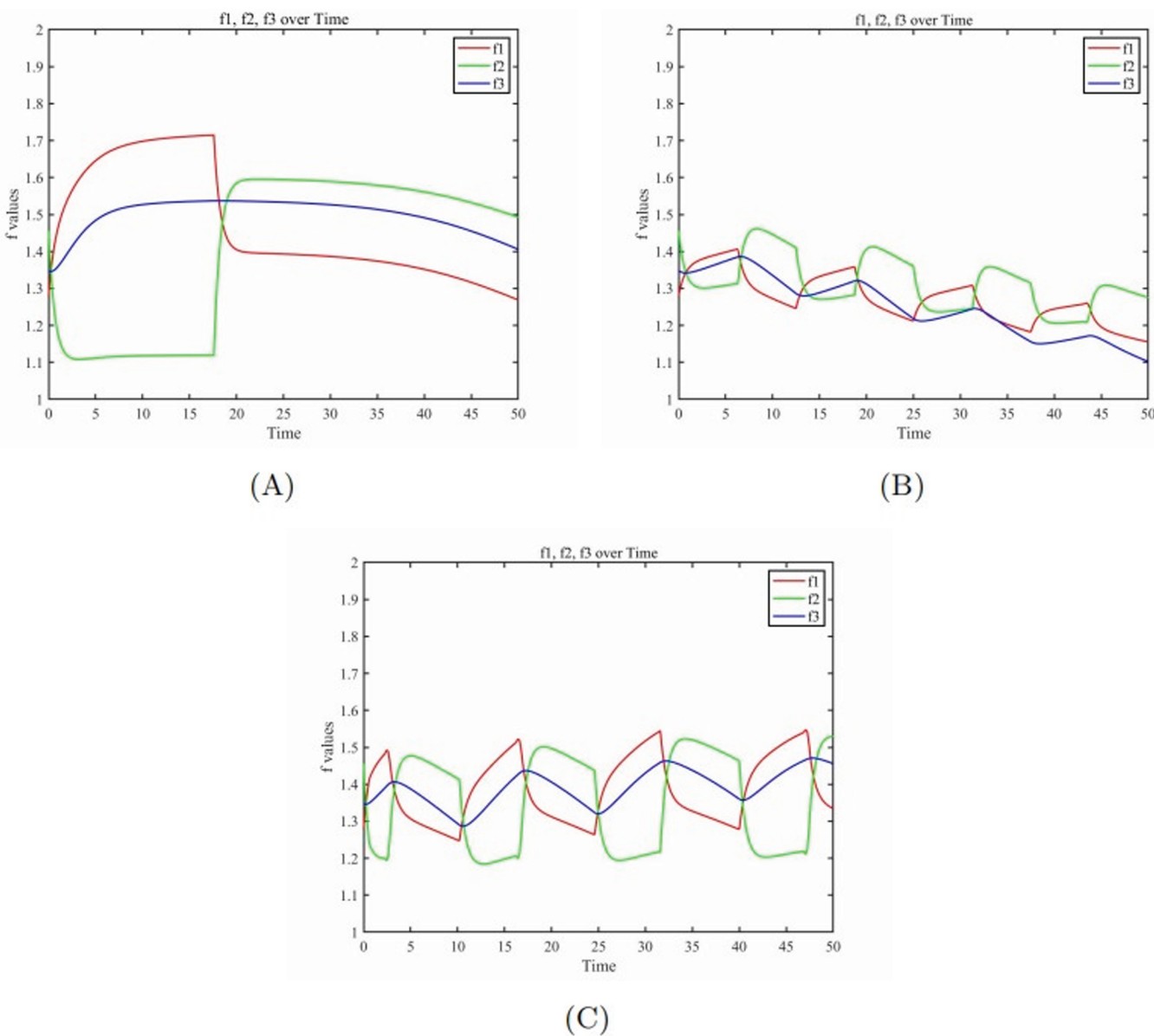

**Fig 8. Below are the fitness functions for healthy cells, drug-sensitive cells, and drug-resistant cells.** (A) Fitness functions under the MTD treatment scheme. (B) Fitness functions under the metronomic therapy scheme. (C) Fitness functions under the optimal adaptive therapy scheme.

oscillation state. Fig 7(D),(E),(F) compares the changes in cancer cell proportions during the three treatment strategies, illustrating that optimal adaptive therapy maintains healthy cells as the dominant population throughout the treatment process.

Fig 8 describes the changes in fitness functions for each cell subpopulation across the three treatment strategies. The fitness function represents the survival competitiveness of each cell subpopulation. Fig 8 (A) In the MTD treatment scheme, the fitness function of healthy cells shows a higher level compared to the fitness function of tumor cells during the treatment period. However, at the conclusion of the drug administration, healthy cells consistently have

lower fitness compared to the two types of cancer cells. This indicates that, after the treatment ends, the survival ability of healthy cells is lower than that of the two types of cancer cells. Fig 8 (B) In the metronomic therapy scheme, while the fitness level of healthy cells is higher than that of drug-sensitive and drug-resistant cells during the phases of treatment, the fitness level of drug-sensitive cells significantly surpasses that of the other two cell subpopulations during the intervals between treatments. This period's fitness level of drug-sensitive cells is also higher than the fitness level of healthy cells during the treatment periods. This is why, by the end of the treatment, healthy cells are gradually replaced by drug-sensitive cells. Fig 8 (C) The optimal adaptive therapy avoids the competitive release between drug-sensitive and drug-resistant cells. Unlike in the metronomic therapy scheme, the fitness level of healthy cells during treatment periods is higher than that of drug-sensitive cells during the intervals between treatments. Thus, the average fitness function level of healthy cells is higher than that of tumor cells throughout the entire treatment period. This ensures that healthy cells have good competitive survival ability under the optimal adaptive therapy strategy.

## Personalized treatment

In actual clinical cancer treatment, different cancer patients and various cancer types often have complex and diverse conditions. Therefore, we can tailor treatment plans according to the needs of different patient groups, developing suitable therapies. Here, we introduce the optimal adaptive therapy strategies for two types of patient groups:

**Low tumor burden group.** For patients with relatively low resistance, drug toxicity, and tumor burden tolerance, a low drug dosage treatment is adopted. Keeping the tumor population proportion within a relatively low tumor burden range is key to prolonging patient survival and maintaining a relatively high quality of life during treatment. Considering an initial proportion of healthy cells and tumor cells of $x_1, x_2, x_3 = (0.4, 0.5, 0.1)$, we set the patient's maximum drug tolerance $c_{max} = 0.6$ and the maximum tumor burden $T_{max} = 0.4$. Through numerical simulation, we obtained the optimal drug dosages ($m_1 = 0.75, m_2 = 0.745, m_3 = 0.745, m_4 = 0.745$) and the optimal treatment on/off periods ($t_1 = 0.6, t_2 = 5.6, t_3 = 12, t_4 = 16.4, t_5 = 22.8, t_6 = 27.6, t_7 = 34, t_8 = 39.6$). Fig 9 (A) and (B) show that during the treatment, the proportions of healthy cells and tumor cells remain in a low-value oscillation state, proving the stability and effectiveness of this strategy. Figs 9(C) and 9(D) depict the changes in drug dosage and concentration within the patient's system during the optimal on/off treatment periods. These figures further validate the effectiveness of this personalized optimal adaptive therapy strategy. Based on the above drug dosage and tumor burden adjustments, this personalized optimal adaptive therapy strategy not only reduces the toxic effects of chemotherapy to a certain extent but also improves the patient's quality of life during treatment.

**High tumor burden group.** We design personalized treatment plans for patients with the same initial conditions as those in the personalized treatment group with low tumor burden.Given that patients with a high tumor burden have better tolerance to drug toxicity, we set the maximum drug tolerance concentration for patients to $c_{max} = 0.79$ and the maximum tumor burden to $d_{max} = 0.5$. Since these patients exhibit good tolerance to drug toxicity, we use the maximum drug dosage for treatment. Through numerical simulations, we obtained the optimal treatment on/off times ($t_1 = 2.4, t_2 = 4.2, t_3 = 14.2, t_4 = 18.6, t_5 = 28.6, t_6 = 33.2, t_7 = 43.2, t_8 = 48.2$). Fig 10 (A) and 10 (B) illustrate that under the conditions of high drug tolerance and high tumor burden, healthy cells fluctuate within the range of 50% to 75%. Due to the patient's mechanism of high drug toxicity tolerance and high tumor burden capacity, tumor cell have more room for maneuver. Fig 10 (C) is based on the patient's drug tolerance

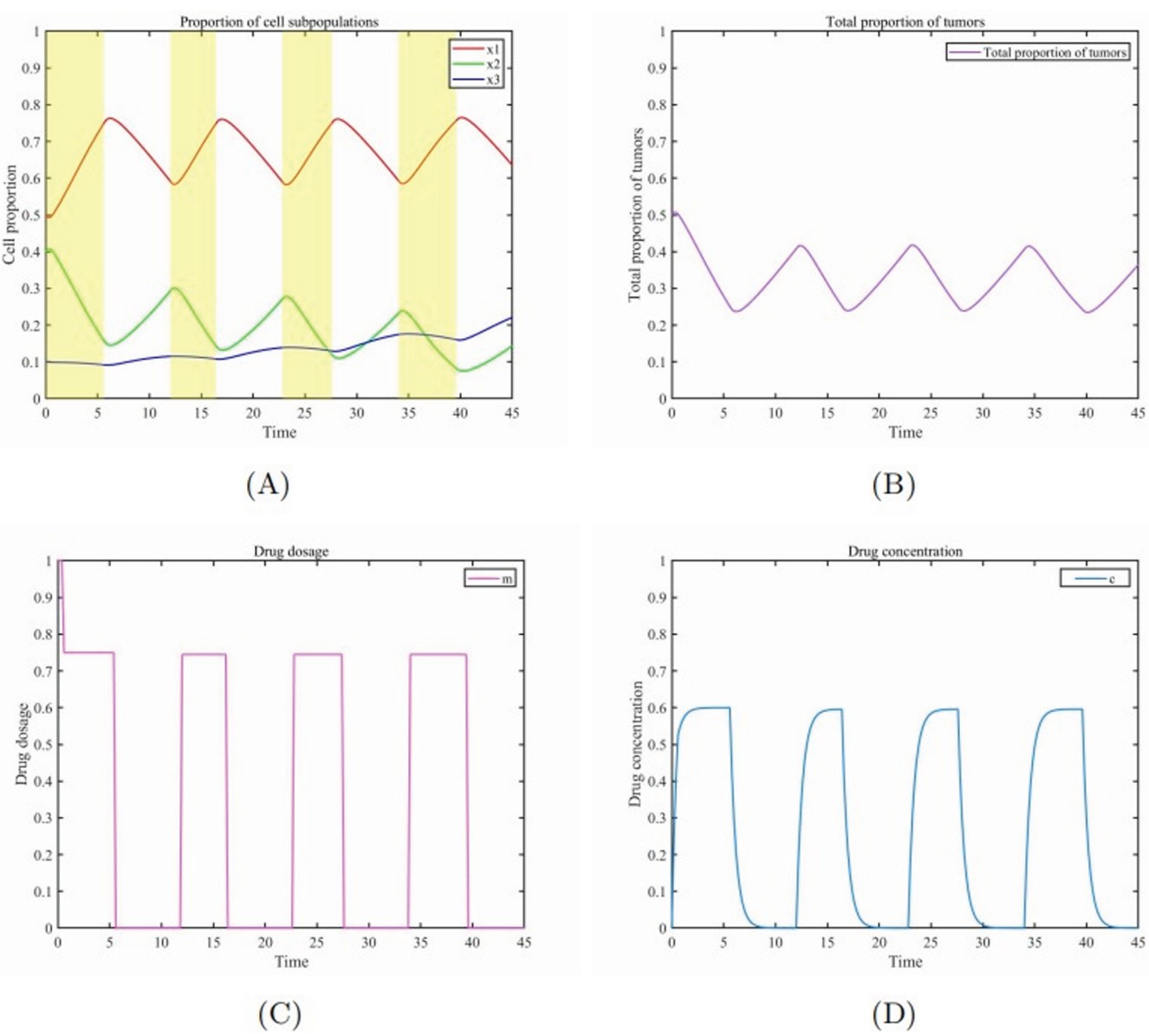

**Fig 9. Personalized treatment for low tumor burden cases.** (A)Temporal evolution of the proportions of various cell subpopulations. (B) Temporal evolution of the overall proportion of cancer cells. (C) Drug dosage. (D) Changes in drug concentration.

and tumor burden capacity to implement the drug administration plan. Fig 10 (D) shows the changes in drug concentration in the patient's body during treatment.

## Discussion

Despite the significant role of evolutionary research in cancer treatment, there are very few studies that have incorporated evolutionary models into clinical cancer treatment. The emergence of adaptive therapy [29,30], a new treatment approach, has been applied to various cancer studies. However, accurately determining the dosage and duration of a fixed treatment period remains a major therapeutic challenge.

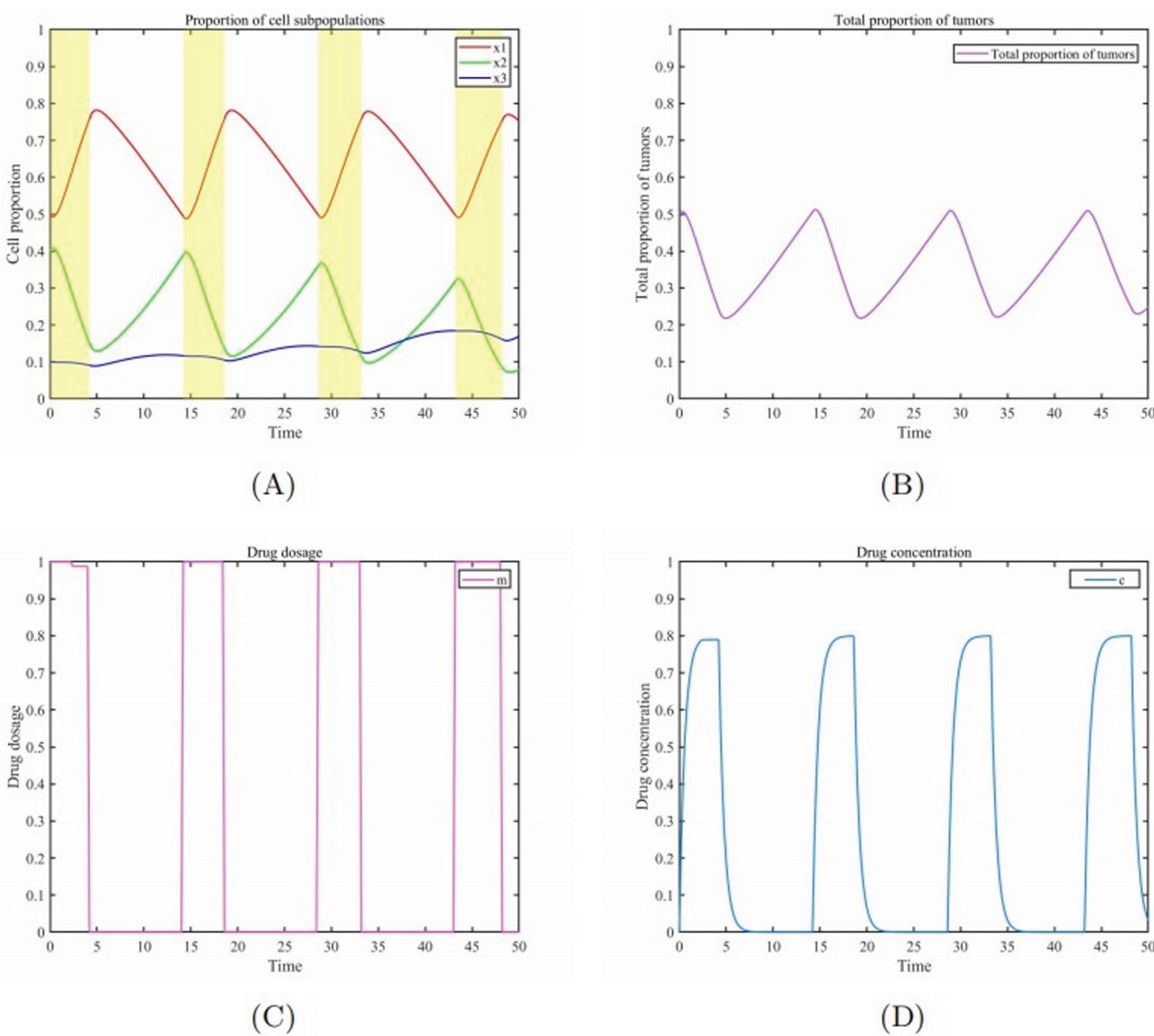

**Fig 10. Personalized treatment for high tumor burden cases.** (A)Temporal evolution of the proportions of various cell subpopulations. (B) Temporal evolution of the overall proportion of cancer cells. (C) Drug dosage. (D) Changes in drug concentration.

In this work, we integrated pharmacokinetics into evolutionary game theory to better characterize the accumulation and clearance of chemotherapy drugs in the human body during treatment, thus enabling the implementation of more precise drug treatment plans. Building on this model, we proposed an optimal control problem constrained by the highest permissible drug concentration and the peak tumor burden state, with the goal of minimizing drug toxicity and the proportion of tumor cells by the conclusion of treatment. Using the Pontryagin minimum principle, we derived the optimal control structure to design the optimal adaptive therapy strategy, optimizing drug dosage and treatment on/off times.

In the numerical simulation phase, we systematically compared the performance differences between the optimal adaptive therapy scheme and other therapy schemes (as shown in Figs 7 and 8) and conducted a detailed analysis from multiple perspectives. First, the maximum tolerated dose (MTD) therapy scheme exhibited significant limitations. This scheme failed to adequately consider the competitive balance mechanism among different subpopulations of cancer cells, leading to the disruption of the dynamic equilibrium between drug-sensitive and drug-resistant cells during treatment. Specifically, under MTD therapy, drug-sensitive cells were largely eliminated due to the high drug dosage, while drug-resistant cells, which were previously suppressed by drug-sensitive cells, lost their competitive constraints. As a result, these drug-resistant cells proliferated rapidly and eventually dominated the tumor population (as shown in Fig 7(A) and (B)). Although this "one-size-fits-all" high-intensity treatment approach may yield short-term therapeutic benefits, its neglect of tumor heterogeneity makes it unsuitable for achieving long-term control objectives.

Second, the rhythmic therapy scheme, while introducing temporal factors by administering drugs at fixed intervals to mimic tumor proliferation patterns, still had notable limitations. This scheme assumed that both tumor proliferation cycles and drug efficacy time windows were fixed parameters, an assumption that deviates significantly from clinical reality. In actuality, tumor growth rates, intercellular competition dynamics, and pharmacokinetic characteristics can vary dynamically due to individual differences and disease progression stages. Therefore, implementing drug administration at fixed time intervals often results in a mismatch between drug delivery timing and the tumor's actual state (as shown in Fig 7(C) and (D)). This imprecise treatment approach not only fails to effectively control tumor growth but may also lead to suboptimal therapeutic outcomes or accelerated resistance development due to over-treatment or under-treatment.

Furthermore, although West's proposed adaptive therapy scheme theoretically achieved closed-loop tumor control by maintaining tumor volume within a stable range (i.e., fluctuating between the initial tumor volume and a preset threshold), it still had critical shortcomings in clinical application. Specifically, West's scheme focused excessively on quantitative tumor volume control while neglecting patients' quality of life, a crucial consideration. In real clinical scenarios, patients' systemic conditions are not static: prolonged exposure to high tumor burdens can lead to immune dysfunction, nutritional deterioration, and organ damage. Thus, a treatment approach that prioritizes tumor volume control without considering patients' overall health status does not align with the ultimate goals of clinical practice.

In contrast, the optimal adaptive therapy scheme demonstrated significant advantages in several aspects: first, by dynamically optimizing treatment timing and drug dosages, this scheme effectively suppressed tumor growth while minimizing damage to healthy tissues. This precise treatment strategy helped maintain a high proportion of surviving healthy cell populations. Second, the optimal adaptive scheme maintained a robust competitive relationship between drug-sensitive and drug-resistant cells. By rationally regulating their population ratios, it effectively delayed the explosive growth of drug-resistant cell populations (as shown in Fig 7(E) and (F)). Third, this scheme fully considered patient-specific differences and tolerance levels. On one hand, it ensured patients' tolerance to treatment-related toxicities; on the other hand, it also accounted for patients' capacity to tolerate tumor burdens. Lastly, unlike West's scheme, the optimal adaptive scheme not only focused on achieving tumor control objectives but also placed special emphasis on patients' quality of life. By avoiding overtreatment and precisely controlling tumor burden, this scheme could sustain patients' healthy states and quality of life over extended periods. Additionally, building upon the optimal adaptive therapy scheme, we further developed personalized treatment plans tailored to different patient populations.

In summary, the optimal adaptive therapy scheme offers precisely tailored drug dosages and treatment schedules for patients with varying conditions, providing a new therapeutic framework for clinical practice. Additionally, we can enhance this model's applicability by integrating combination therapies involving multiple drugs into the evolutionary game model. However, implementing such treatment plans depends on having an accurate understanding of the system's current state. Current clinical biopsy techniques [31,32] are not sufficiently advanced; traditional tissue biopsies require surgical or needle procedures to obtain tissue samples, which are invasive and carry risks such as infection and bleeding. Moreover, tissue biopsies only provide information about specific tumor sites and may not reflect the condition of the entire tumor or other metastatic locations. Overcoming these limitations in detection technology is a significant challenge for the implementation of adaptive therapy schemes.

## Acknowledgments

The authors would like to thank the anonymous reviewers for their valuable comments and suggestions.

## Author contributions

**Conceptualization:** Zhiqing Li.

**Formal analysis:** Zhiqing Li.

**Funding acquisition:** Xuewen Tan.

**Methodology:** Zhiqing Li.

**Software:** Yangtao Yu.

**Supervision:** Xuewen Tan.

**Validation:** Zhiqing Li, Yangtao Yu.

**Visualization:** Yangtao Yu.

**Writing – original draft:** Zhiqing Li.

**Writing – review & editing:** Xuewen Tan, Yangtao Yu.

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
