## [Decision Letter · Decision Letter 0]

21 Jan 2025

PONE-D-24-40566Optimal Adaptive Cancer Therapy Based on Evolutionary Game TheoryPLOS ONE

Dear Dr. Li,

Thank you for submitting your manuscript to PLOS ONE. After careful consideration, we feel that it has merit but does not fully meet PLOS ONE’s publication criteria as it currently stands. Therefore, we invite you to submit a revised version of the manuscript that addresses the points raised during the review process.

We look forward to receiving your revised manuscript.

Kind regards,

Nikos Kavallaris, Ph.D

Academic Editor

PLOS ONE

“This work was supported by the National Natural Science Foundation of China (Nos. 11361104, 12261104), the Youth Talent Program of Xingdian Talent Support Plan (XDYC-QNRC 2022-0514), the Yunnan Provincial Basic Research Program Project (No. 202301AT070016, No. 202401AT070036), the yunnan Province International Joint Laboratory for Intelligent Integration and Application of Ethnic Multilingualism (202403AP140014).”

5. We note that you have indicated that there are restrictions to data sharing for this study. PLOS only allows data to be available upon request if there are legal or ethical restrictions on sharing data publicly. For more information on unacceptable data access restrictions, please see http://journals.plos.org/plosone/s/data-availability#loc-unacceptable-data-access-restrictions.

Reviewers' comments:

Reviewer's Responses to Questions

**Comments to the Author**

1. Is the manuscript technically sound, and do the data support the conclusions?

Reviewer #1: Yes

Reviewer #2: Yes

2. Has the statistical analysis been performed appropriately and rigorously? 

Reviewer #1: Yes

Reviewer #2: Yes

3. Have the authors made all data underlying the findings in their manuscript fully available?

Reviewer #1: Yes

Reviewer #2: Yes

4. Is the manuscript presented in an intelligible fashion and written in standard English?

Reviewer #1: Yes

Reviewer #2: Yes

5. Review Comments to the Author

Reviewer #1: Title: Optimal Adaptive Cancer Therapy Based on Evolutionary Game Theory

Summary:

Authors incorporate pharmacokinetics into a cancer evolutionary game theory model of adaptive cancer therapy. The model framework builds upon a previous game theoretic model of adaptive therapy but goes beyond previous work by 1) incorporating drug decay dynamics and 2) solving an optimal control problem. The work is thus novel and a very useful and welcome addition to the literature in this field.

Major comments

1. Firstly, can J in equation 10 be weighted between the two terms? I’m concerned that the first term is by definition between 0 and 1, while the second term can be infinite as time tF goes to infinity – does this essentially negate the effect of the first term in the control problem?

2. The most interesting part of the manuscript to me is found in lines 308-310 where the optimal treatment schedule maintains the same value for m1 m2 m3 and m4. This is remarkably reminiscent to the “dose-skipping” adaptive strategy proposed in the original series of papers. In many follow ups, the goal has been to compare dose-skipping versus dose modulation adaptive strategies. I think this should be noted in the discussion, with a comment on how the optimal strategy is a dose-skipping one. For example see this paper: https://www.cell.com/cell-systems/abstract/S2405-4712(24)00118-2 (there are many other examples as well).

Minor comments:

1. Missing reference in line 246

2. I am curious to know what “environmental factors” referenced in line 247 means. It’s not important for the paper as long as the payoff matrix criteria in lines 80 - 82 are satisfied, so maybe this comment can be removed.

Reviewer #2: Review report on Optimal Adaptive Cancer Therapy Based on Evolutionary Game Theory by Zhiging Li, Xuewen Tan and Yangtao Yu

In this paper, authors adopted an evolutionary game theory approach combined with optimal control theory to model cancer growth and chemotherapeutic treatment mechanism. Also incorporated pharmacokinetics of the drug into the modeling frame work. They proposed an optimal adaptive therapy strategy. Existence of the solution of optimal control problem is derived and characterization of it using Pontryagin’s minimum principle is established. They conducted a numerical simulation to substantiate the theoretical proposal and compared with other standard approaches.

The theoretical derivation seems to be correct and from numerical simulation we can see some advantage of the proposed model compared to other models but not completely eradicate the tumor population. The concern of the reviewer is the following: Will game theory model can actually deal with practical cases as we need to deal with human beings? Game theory has applied to other applicable areas but we need to be little bit cautious while applying in human beings. Even other types of mathematical models have limitations while applying to real-time situations as mathematical models basically works based on assumptions. Since, cancer is a killer disease, no harm in studying a new approach for solving the problem and proposing treatment strategies. Any new approach in studying the problem should always welcome. In that perspective, although it has certain shortfalls, this reviewer would like to recommend the paper for publication.

Correction: The authors should mention the missing reference paper on page number 10, line 246 regarding the parameters used for simulation.

6. PLOS authors have the option to publish the peer review history of their article (what does this mean?). If published, this will include your full peer review and any attached files.

Reviewer #1: No

Reviewer #2: No

---

## [Author Response · Author response to Decision Letter 1]

19 Feb 2025

Reviewer 1

Major Comments:

Comment 1:

• Reviewer’s Comment: Can a weighting factor be introduced between the two terms in Equation 10? When tf approaches infinity, does the first term in the objective function lose its role in the control problem?

• Authors’ Response: The issue you raised is crucial, and we fully agree. In the original objective functional, the integral term may dominate as time increases, potentially diminishing the role of the first term. To address this, we have introduced a weighting coefficient in the revised version, modifying the functional as:

min⁡J(x,m)=λψ(T(t_f))+∫_0^(t_f)ηm(t)dt

Here, the weighting coefficient λand η are selected based on practical requirements to balance the contributions of both terms. This adjustment prevents the integral term from dominating over long time horizons while preserving the regulatory role of the terminal termψ(T(t_f)).

• Revision Location: Lines 113–114, Lines 149–151�Lines 154–155,Equations 10, 11, 17,24 and 27.

Comment 2:

• Reviewer’s Comment: Please elaborate in the Discussion section on why the optimized adaptive therapy strategy is superior to other treatment strategies.

• Authors’ Response: Thank you for the suggestion. We have added a discussion comparing the advantages of our optimal adaptive therapy strategy with other methods, including those proposed by West et al., citing relevant literature.

• Revision Location: Lines 444–496.

Minor Comments:

Comment 1:

• Reviewer’s Comment: A citation is missing in Line 246.

• Authors’ Response: Thank you for pointing this out. We have added the appropriate reference.

• Revision Location: Line 247.

Comment 2:

• Reviewer’s Comment: Clarify the meaning of "environmental factors" in Line 247.

• Authors’ Response: "Environmental factors" refer to patient-specific differences influencing cancer cell dynamics, such as genetic mutations, organ function, immune status, and drug metabolism. These factors lead to heterogeneous responses to therapy, as validated by clinical data from Virginia et al. (https://doi.org/10.1101/2021.10.29.466444). In our study, the competition matrix was adjusted to reflect moderate drug resistance, allowing clearer demonstration of adaptive therapy efficacy without altering the fundamental competitive relationships.

Reviewer 2

Comment 1:

• Reviewer’s Comment: While numerical simulations show the superiority of adaptive therapy, it fails to fully eradicate tumors. Moreover, can the game-theoretic model be safely applied to real patients?

• Authors’ Response: Adaptive therapy aims to maintain a balance between healthy and cancerous cells (sensitive and resistant) to prolong survival, rather than complete eradication. We acknowledge the need for caution when translating theoretical models to clinical practice. However, recent studies (e.g., Virginia et al., https://doi.org/10.1101/2021.10.29.466444) demonstrate promising clinical validation of game-theoretic models. Future work will integrate more biological mechanisms and deep learning to enhance predictive accuracy.

Comment 2:

• Reviewer’s Comment: A citation is missing in Line 246.

• Authors’ Response: Thank you for highlighting this. The reference has been added.

• Revision Location: Line 247.

---

## [Decision Letter · Decision Letter 1]

23 Feb 2025

Optimal Adaptive Cancer Therapy Based on Evolutionary Game Theory

PONE-D-24-40566R1

Dear Dr. Li,

We’re pleased to inform you that your manuscript has been judged scientifically suitable for publication and will be formally accepted for publication once it meets all outstanding technical requirements.

Kind regards,

Nikos Kavallaris, Ph.D

Academic Editor

PLOS ONE

Additional Editor Comments (optional):

Reviewers' comments:

Reviewer's Responses to Questions

**Comments to the Author**

1. If the authors have adequately addressed your comments raised in a previous round of review and you feel that this manuscript is now acceptable for publication, you may indicate that here to bypass the “Comments to the Author” section, enter your conflict of interest statement in the “Confidential to Editor” section, and submit your "Accept" recommendation.

Reviewer #1: All comments have been addressed

2. Is the manuscript technically sound, and do the data support the conclusions?

Reviewer #1: Yes

3. Has the statistical analysis been performed appropriately and rigorously? 

Reviewer #1: Yes

4. Have the authors made all data underlying the findings in their manuscript fully available?

Reviewer #1: Yes

5. Is the manuscript presented in an intelligible fashion and written in standard English?

Reviewer #1: Yes

6. Review Comments to the Author

Reviewer #1: (No Response)

7. PLOS authors have the option to publish the peer review history of their article (what does this mean?). If published, this will include your full peer review and any attached files.

Reviewer #1: No

---

## [Editor Report · Acceptance letter]

PONE-D-24-40566R1

PLOS ONE

Dear Dr. Li,

I'm pleased to inform you that your manuscript has been deemed suitable for publication in PLOS ONE. Congratulations! Your manuscript is now being handed over to our production team.

Kind regards,

on behalf of

Dr. Nikos Kavallaris

Academic Editor

PLOS ONE